# A Novel Prediction Model for Malicious Users Detection and Spectrum Sensing Based on Stacking and Deep Learning

**DOI:** 10.3390/s22176477

**Published:** 2022-08-28

**Authors:** Salma Benazzouza, Mohammed Ridouani, Fatima Salahdine, Aawatif Hayar

**Affiliations:** 1RITM Laboratory, CED Engineering Sciences, Hassan II University, Casablanca 20000, Morocco; 2Department of Electrical and Computer Engineering, University of North Carolina at Charlotte, Charlotte, NC 28223, USA

**Keywords:** cognitive radio network, compressive sensing, spectrum sensing, malicious users detection, machine learning, stacking, deep learning, convolutional neural network

## Abstract

Cooperative network is a promising concept for achieving a high-accuracy decision of spectrum sensing in cognitive radio networks. It enables a collaborative exchange of the sensing measurements among the network users to monitor the primary spectrum occupancy. However, the presence of malicious users leads to harmful interferences in the system by transmitting incorrect local sensing observations.To overcome this security related problem and to improve the accuracy decision of spectrum sensing in cooperative cognitive radio networks, we proposed a new approach based on two machine learning solutions. For the first solution, a new stacking model-based malicious users detection is proposed, using two innovative techniques, including chaotic compressive sensing technique-based authentication for feature extraction with a minimum of measurements and an ensemble machine learning technique for users classification. For the second solution, a novel deep learning technique is proposed, using scalogram images as inputs for the primary user spectrum’s classification. The simulation results show the high efficiency of both proposed solutions, where the accuracy of the new stacking model reaches 97% in the presence of 50% of malicious users, while the new scalogram technique-based spectrum sensing is fast and achieves a high probability of detection with a lower number of epochs and a low probability of false alarm.

## 1. Introduction

Cognitive radio is a potential technology that has proven its efficiency since it was introduced by Joseph Milota in 1998. It is designed mainly for efficient radio spectrum allocation using an intelligent system to identify and exploit vacant radio spectrum without interferences [1]. The sophisticated structure of the cognitive radio system has addressed the crucial spectrum-shortage problem in future wireless communications, where the number of subscribers is growing rapidly. In a cognitive radio network (CRN), finding an unused spectrum is performed using the spectrum sensing process [2], which uses nodes known as secondary users (SUs) without legal licences to sense and check the occupation of an optional spectrum used by other nodes known as primary users (PUs) with legal licences.

To achieve high spectrum sensing performances, centralized and distributed cooperative networks are proposed as relevant solutions [3]. In the first network, SUs cooperate and share their sensing data with a fusion center (FC), which initially gathers all the SUs’ sensing observations to then reach an optimal decision on the PUs’ spectrum occupancy. In the second network, SUs cooperate, share the sensing information between them, and make the final decision about PU spectrum occupancy autonomously, without any interaction with an FC. Despite the key benefits of cooperative networks, they are susceptible to potential attacks by malicious users (MUs) causing undesirable interferences between SUs and PUs, and consequently reducing the decision accuracy of the spectrum sensing process [4]. In this paper, we considered an MU as being an unreliable user that decreases the CRN performances for various reasons, including a malicious attacker attempting to harm the system, or a secondary user with a low received signal and uninformative sensing data. Hence, the design of advanced cooperative network techniques are extremely needed, to protect the network against MU threats in the first step, and to deeply explore the spectrum of interest for a reliable decision in a second step.

In the literature, several techniques have been implemented and discussed for either MUs detection or PU spectrum sensing. The conventional techniques are based mainly on a thresholding comparison method, as proposed in [5,6,7,8,9,10]. For MUs detection, the authors in [5] use the box-whisker plot based on a correlation metric to deal with MUs in CRN. The box determines the inferior and superior quartile limits for each node. A given node is declared a normal SU if its data are outside of these two quartile limits; otherwise it is considered as an MU. The authors in [6] proposed an adaptive thresholding method based on the mass values of the nodes. The authors use double thresholds $Th_{1}$ and $Th_{2}$ in the sensing process to classify each node into three classes. A particular node is defined as a legitimate SU if its mass value is greater than $Th_{2}$. Otherwise, if the mass value is less than $Th_{1}$, the node is declared as an MU, and last, if the mass value is between $Th_{1}$ and $Th_{2}$, the node is identified as a doubtful user. The authors in [7] introduce a new technique based on fuzzy logic for reliable SU selection. This technique uses three measuring parameters jointly: SNR, channel quality, and trust factor as inputs to increase the system accuracy. For PU spectrum sensing, the authors in [8] proposed a new dynamic threshold based on a time quite approach for matched filter technique-based spectrum sensing in CRN. The authors in [9] introduced a new spectrum sensing technique for cooperative CRNs using chaotic sensing matrices for signal acquisition, and an adaptive threshold-based energy detection to achieve the final spectrum sensing decision. Another type of PU spectrum sensing technique based on auto-correlation comparison is proposed in [10], where the authors proposed a new Bayesian compressive method for a wideband signal recovery process, followed by an auto-correlation-based detection technique to perform an accurate PU spectrum sensing decision.

Most recently, techniques based on machine learning have been widely developed, overcoming a number of existing limitations of conventional techniques. Machine learning is the most popular field in artificial intelligence, and aims to give machines the ability to self-learn. The most commonly used machine learning techniques are supervised learning [11], which performs a training process based on labeled input samples, and unsupervised learning [12], which performs the training process with unlabeled input samples. For the more complicated tasks, usually deep learning is used, which is a subset of the machine learning field, and consists of layered structures known as artificial neural networks inspired by the human brain [13]. Examples of machine learning-based techniques are proposed in [14,15,16,17,18,19,20,21] and summarized in Table 1. For MUs detection, the authors in [14] implemented a new support vector machine (SVM) algorithm to separate MUs from SUs under a three class hypothesis. They used the energy vector of the nodes cooperating in the network as an input feature. The proposed algorithm is evaluated in terms of accuracy, receiver operating characteristic (ROC), probability of detection ($P_{d}$), and false alarm ($P_{fa}$). In [15], the authors used three individual classifiers, namely logistic regression (LR), k-nearest neighbors (k-NN), and SVM to detect MUs in energy harvested CR-IoT networks. The proposed LR algorithm gives better performance results in terms of accuracy, sum rate, and network lifetime. In [16], the authors presented a machine learning framework using six individual classification models, namely LR, KNN, SVM, Linear Discriminant Analysis (LDA), Decision Tree Classifier (DTC), and Gaussian Naive Bayes (NB) to evaluate a PU emulation attack in CRN in terms of accuracy, recall, precision, and F-score. The authors used a new pattern described as the link signature method for features extraction. In the same context, the authors in [17] proposed a new algorithm based on the sparse recovery technique to distinguish between a reliable PU and attackers. Then, a machine learning operation was performed for classification and decision-making. The proposed technique was evaluated in terms of ROC and area under ROC metrics. For PU spectrum sensing, the authors in [18] performed a comparative study of different machine learning techniques; namely, random forest, SVM, KNN, DCT, NB, and LR, using the energy statistic as the input feature vector, to investigate their efficiency for the spectrum sensing process. The authors in [19] proposed an adaptive cooperative sensing technique using a weighted ensemble model. This model used three different feature vectors: energy feature, wavelet feature, and SNR feature. Each feature vector is trained by a different machine learning technique; namely, DTC, SVM, and KNN, respectively. At last, the outputs of the three techniques are combined using a weighted method to decide on the occupancy of the PU spectrum. In the same context, the authors in [20] proposed a novel convolution neural network (CNN)-based solution to perform PU spectrum sensing using the basic features: cyclostationary and energy feature. The authors in [21] investigated an automatic features extraction method based on a deep learning technique. This technique combined a CNN, long short-term memory (LSTM), and fully connected neural network together, to reach high PU spectrum sensing performances.

However, all the above-discussed techniques were firstly performed using either MUs detection or PU spectrum sensing. Secondly, they depended mostly on the pre-knowledge of the recent activity of the nodes, a requirement that is not always available due to the possibility of dynamic changes in the parameters of an MU. Thirdly, the proposed machine learning-based techniques commonly use one-dimensional signal analysis (in the time or frequency domain), which prevents a full extraction of hidden features. To deal with these different issues, we have proposed a comprehensive approach considering both the MUs detection, as well as the PU spectrum sensing. Our approach involves two techniques, namely stacking model-based authentication and deep learning-based spectrum sensing. The first technique was inspired from our previous work [22]; it aims at selecting reliable SUs and eliminating MUs using machine learning techniques, and without prior knowledge of the activity of the users. The proposed stacking model-based authentication involves a chaotic compressive sensing process used for gathering the features and building our dataset, followed by a stacking learning process used for making an accurate decision regarding node reliability. In addition, we have introduced new metrics to evaluate our stacking model, namely, Hamming loss score, Jaccard score, authentication rate, and intrusion rate. The second technique aims at performing PU spectrum sensing with a high probability of detection via the time-frequency analysis of the received SU signal. In this technique, we performed a comparative study between various deep learning models based on scalogram images to decide on the occupancy of the PU spectrum.

The remainder of this paper is organized as follows: Section 2 presents the methodology involved in the proposed approach. Section 3 presents the simulation results of the proposed techniques based on different metrics. Finally, a conclusion is presented at the end.

## 2. Materials and Methods

In this section, we proposed a new approach based on two machine learning solutions to improve cooperative CRNs by considering two main aspects: (i) the confidentiality of the sensed data and (ii) the accuracy of the spectrum sensing decision. For the first aspect, a new machine learning solution-based stacking model is proposed for MUs detection using two main techniques, including chaotic compressive sensing technique-based authentication and the stacking machine learning technique. For the second aspect, a new deep learning solution based on scalogram plots is proposed for the PU spectrum’s classification. Figure 1 represents the main blocks of the proposed approach, where an authentication process is introduced before performing the spectrum sensing process. The first block aims at avoiding the harmful interferences that can be caused by the presence of the MUs, which directly affects the detection performance of a cooperative CRN. The second block performs an intelligent and secure spectrum sensing with a high probability of detection.

This section is organized into two sub-sections to introduce our approach. The first sub-section outlines the detailed solution adopted to automatically reject the MUs into the network using ensemble machine learning. The second sub-section discusses the comparative analysis performed to select the most appropriate deep learning model for the spectrum sensing process.

### 2.1. A Stacking Model-Based Malicious Users Detection

We considered a centralized cooperative CRN with I nodes classified into Z normal nodes (SUs) and L abnormal nodes (MUs), which cooperate with an FC to perform the spectrum sensing process. Before reaching the final decision about the spectrum occupation, all CRN users, including SUs and MUs, initially performed a double authentication process based on compressive sensing technique [23,24], combined with a machine learning approach to confirm their reliability. Then, once the corresponding FC has successfully completed its classification, each user identified as a trusted user is automatically allowed to start the spectrum sensing process. Figure 2 presents the main processes of the new proposed technique, namely the authentication process and the machine learning process. In the authentication process, we assumed that the reliable nodes and the FC share two secure pre-shared matrices, $M_{1}$ and $M_{2}$, which are generated using a Chebyshev map [25]. These chaotic sensing matrices are used as a security key to distinguish between normal and abnormal nodes [26]. The proposed double authentication process involves two steps. The first step performed is the compression of a random sparse signal $m_{s}$ at the FC, followed by a recovery of the same signal at the node using the same sensing matrix $M_{1}$. The second performed step is a compression of the recovered signal $m_{sr}$ at the node followed by a second recovery at FC using the same sensing matrix $M_{2}$. More details regarding the double authentication process are described in our previous work [22]. In the machine learning process, three main steps are performed by the FC, namely feature selection, classification modeling, and node reliability decision. At the feature selection step, the FC gathered different features based on the received compressive vectors from all nodes ($SU_{z}$ and $MU_{l}$) to build its dataset. At the classification-modeling step, the model is generated, trained, and tested using individual or ensemble machine learning algorithms [27]. At the decision step, the FC is able to recognize whether the node is a SU or MU, and finally decides to accept or reject it. The following sections describe in detail the approach adopted for the machine learning process.

#### 2.1.1. Features Selection

Usually, the aim of the authentication process is the ability of the FC to distinguish between an SU and an MU by comparing the original signal $m_{s}$ generated at the beginning of the request, with the signal reconstructed based on the measurement vector $Y_{2}$ received from each node. The final recovered signal of $node_{i}$ can be described as follows,
(1)$$r_{i}=argmin|m_{sr}^{i}{|}_{1}\;\mathrm{subject}\;\;\;\mathrm{to}\;\;Y_{2}^{i}=M_{2}\times m_{sr}^{i}+e^{i}$$
where *r* is the recovered signal of $m_{sr}$, ${∥.∥}_{1}$ is the norm 1, *Y* denotes the compressive measurement vector, $M_{2}$ presents the chaotic sensing matrix, and *e* is the additive Gaussian noise. Consequently, different features can be selected and extracted by comparing the original signal $m_{s}$ and the recovered signal $r_{i}$ described by Equation (Equation 1). In this paper, we used five specific and relevant features, which can help to classify the nodes into two categories (SU and MU). These features are: recovery error, spikes error rate, sparsity error, magnitude-squared coherence, and mean squared error [28].

Recovery error:Also denoted by reconstruction error, this is a metric used to measure the error level between the original signal and the recovered signal according to the distance between the two signals. It can be formulated mathematically as
(2)$$R_{e}=\frac{∥m_{s}-r∥}{∥m_{s}∥}$$
where $R_{e}$ is the recovery error, *m* is the original signal, and *r* is the recovered signal.Spikes error rate (SER):It estimates the error rate of a recovered signal in terms of its spikes. It is calculated as a function of missed and false spikes. It can be described as follows,
(3)$$SER=\frac{N_{fs}+N_{ms}}{N}$$
where $N_{fs}$ and $N_{ms}$ are the number of false and missed recovered spikes, respectively, and N is the length of the original signal.Sparsity error:The sparsity of a signal defines the number of significant samples included in a signal. This characteristic takes a crucial function in the process of a compressive sensing technique to achieve a perfect recovery of a sparse signal. Thus, the sparsity error metric can be defined as the rate of failure between the original signal and the recovery signal in terms of their sparsity level. The sparsity error of a recovered signal r can be expressed as follows,
(4)$$k_{error}=\frac{K_{r}}{K_{m}}$$
where $K_{r}$ denotes the sparsity level of the recovered signal *r* and $K_{m}$ denotes the sparsity level of the original signal *m*.Magnitude squared coherence (MSC):It evaluates the similarities of two signals in terms of frequency. It uses a linear model to determine at which degree a signal can be predicted from another signal. The MSC between two signals m and r is a real-valued function that can be defined as
(5)$$C_{mr}\left(f\right)=\frac{|G_{mr}\left(f\right){|}^{2}}{G_{mm}\left(f\right)G_{rr}\left(f\right)}$$
where $G_{mr}\left(f\right)$ denotes the cross-spectral density between *m* and *r*, and $G_{mm}\left(f\right)$ and $G_{rr}$ denote the auto-spectral density of *m* and *r*, respectively.Mean squared error (MSE):It aims at computing the squared error based on the measures of the square of each element of the error signal resulting from the difference between two signals. It measures the extent to which a reconstructed signal and an original signal are different. It also usually used as a performance metric for predictive modeling. The MSE between two signals m and r can be expressed as
(6)$$MSE=\frac{1}{N}\sum_{i=1}^{N}{\left(m(N)-r(N)\right)}^{2}$$
where *m* is the original signal, *r* is the recovered signal, and N denotes the number of samples.

#### 2.1.2. Classification Modeling

Malicious users detection can be represented as a binary classification problem with a dataset D = (X, y) composed of the vector X = ($X_{1}$, $X_{2}$, $X_{3}$, $X_{4}$, $X_{5}$) representing the five input features of a $node_{i}$, and the vector y representing the target feature (or label) of each node, with y = 0 if the node is SU, or y = 1 if the node is MU. To mitigate this classification problem, a stacking model-based ensemble-learning technique is used to achieve more stable and accurate predictions than with individual learning. Stacking or Stacked Generalization is an advanced technique that aims at combining the prediction results of multiple individual learning models to take the final predictions [29]. The stacking architecture includes two or more base models, commonly denoted by level-0 models, and a meta-model denoted by a level-1 model [30], as shown in Figure 3. In level-0, different classification models are trained using the complete training set. Then, in level-1, a single classification model is trained on the outputs of level-0 to best combine the predictions and to take final accurate decisions.

To provide an appropriate stacking with high performances, we used three varied ranges of base models, namely Support Vector Machine (SVM) [31], Logistic Regression (LR) [32], and Gaussian Naïve Bayes (NB) [33]. SVM is one of the most robust supervised algorithms used for solving both regression and classification problems. It performs an accurate classification using a hyperplane that splits two different classes by maximizing the margin between them. LR is a linear technique, usually used for predicting binary classification outputs. For this type of classification, the LR model is mainly based on the logistic function, also called the sigmoid function, with the particularity to be always within the range of 0 and 1. In this work, we used the LR algorithm for both the base model in 0-level and the meta-model in 1-level. NB is a statistical model used for classification problems, with its prediction outputs based on Bayes’ theorem. For our case, we used Gaussian Naive Bayes, since our input features are real-valued. Thus, the proposed stacking Algorithm 1 can be summarized as follows.
**Algorithm 1** Stacking model**Input**: Training data D = ($X_{i}$, $y_{i}$)Level-0 classifiers $C_{1}$, $C_{2}$,$C_{3}$Level-1 classifier C**Output**: Final prediction P**Step 1**: Train level-0 classifiersfor t = 1 to 3 do      $\;\;\;\;\;\;\;\;p_{t}$ = $C_{t}$(D) % train a base model $C_{t}$ based on Dend**Step 2**: Generate new inputs to meta-modelfor i = 1 to m do      $\;\;\;\;\;\;\;\;D^{new}$ = ($X_{i}^{new}$, $y_{i}$) with      $\;\;\;\;\;\;\;\;X_{i}^{new}$ = ($(p_{1}\left(X_{i}\right)$,$p_{2}\left(X_{i}\right)$, $p_{3}\left(X_{i}\right)$)end**Step 3**: Train level-1 classifier$p^{new}$ = C($D^{new}$) % train a meta model C based on $D^{new}$**Step 4**: Take final prediction PP(X) = $p^{new}$ ( $p_{1}\left(X\right)$,$p_{2}\left(X\right)$,$p_{3}\left(X\right)$)

### 2.2. Scalogram-Based CNN Models for Spectrum Sensing

We considered a secure cooperative CRN with a frequency band B = [$f_{n}$ Hz to $f_{m}$ Hz] of interest. The aim task of each SU cooperating in the network is the detection of the occupancy of the band B by PU with a high probability of detection and accuracy. Let $S_{f}$ be a set of frequencies being used by PU, and then the band is considered as occupied if $S_{f}$⊂ B, otherwise, it is considered vacant. To perform the spectrum sensing process, the traditional technique involves three steps [34]. The first step aims at defining a test statistic, which varies according to the technique applied. For instance, energy-based detection takes the sum energy of the received signal as a test statistic [35]. The second step consists of a comparison between the test statistics and a predefined threshold to take a final decision as a last step. However, this type of technique processes the received signal with only one dimension (1D) (either time or frequency), which can reduce the reliability of the spectral analysis, particularly in the presence of intense noises. Furthermore, these techniques have to meet a certain number of requirements for efficient sensing (i.e., the energy-based detection technique requires pre-knowledge of the noise variance).

An alternative to overcoming these issues is to formulate the spectrum sensing process as an image classification problem with a binary hypotheses ($H_{0}$: vacant band or $H_{1}$: occupied band) using deep learning, especially with CNN models for automatic features extraction. In other words, the received signal can be processed as an image of 2D (time-frequency) for better spectral analysis. The scalogram image is an adequate solution for visualizing this time-frequency representation of a signal. It displays the variation of the absolute value of the continuous wavelet transform (CWT) of a signal over time and frequency. In addition, the scalogram provides an accurate frequency localization for long-duration signals with low frequency, as well as a good temporal localization for high-frequency and short-duration signals. Regarding the hardware aspect, coherent detection techniques such as energy-based detection (using in-phase and quadrature (I/Q) data) require hardware that is more complex than that used for the processing of spectrogram/scalogram images [36]. Figure 4 presents examples of a scalogram of different cases.

A scalogram-based CNN models technique for spectrum sensing involves two main processes. The first process aims at transforming the received signal (1D) to a scalogram (2D) using CWT. The second process operates the training of several CNN models with the scalogram images obtained in the first process as the input, as shown in Figure 5.

#### 2.2.1. Continuous Wavelet Transform

The time-frequency analysis of signals is a crucial aspect in signal processing. It aims at accurately localizing the frequency component of a signal as function of time; i.e., instead of analyzing the signal with one dimension (time or frequency), the analysis is performed using two dimensions (time-frequency). The most commonly used techniques to perform time-frequency analysis are the short-time Fourier transform (STFT) [37] and the continuous wavelet transform (CWT) [38]. STFT is an extension of Fourier transforms used to identify the frequency component of the signal over the full time span, using a uniform window size. In practice, the plot usually used for visualizing the STFT results is the spectrogram. In contrast, CWT is a more advanced technique, which is derived from the wavelet transform. Unlike STFT, CWT uses variable window sizes to process highly modulated signals with different resolutions. In other words, CWT evaluates the similarity between a signal and different dilation and shifted forms of a wavelet, which provides the ability to extract more significant features at different scales. To visualize the results of a CWT-based time-frequency analysis, a scalogram plot is usually used. Mathematically, the CWT of a received signal *y*(*t*) can be formulated as follows,
(7)$$Cwt\left(s,p\right)=\frac{1}{\sqrt{s}}\int_{-\infty }^{+\infty }y\left(t\right)\times \psi^{*}\left(\frac{t-p}{s}\right)\;dt$$
where *Cwt*(*s*, *p*) are the wavelet coefficients, s and p are the scale and the position parameters, respectively, ψ(*t*) is the original wavelet known as the mother wavelet, and $\psi_{s,p}$(*t*) is the variation of ψ(*t*) at different scales and positions, known as daughter wavelets. In this study, the Morse wavelet-based CWT are used as the mother wavelet, with Γ = 3, called the symmetry parameter, and a constant b = 60, called the time bandwidth product for normalization [39].

#### 2.2.2. CNN Models

CNN is a robust model of the artificial neural network, also denoted ConvNet, which is is mainly designed for image processing [40,41]. Basically, it includes two main parts, namely features extraction and classification, as shown in Figure 6. The first part involves a sequence of convolutions to split and extract different features from the input images. The second part includes fully connected layers and an output layer with a different number of neurons, depending on the case study for predicting which class the input image belongs to.

To examine the efficiency of the scalogram-based spectrum sensing technique, we performed a comparative study between two approaches based on different CNN models. In the first approach, we explored a basic CNN model without the transfer learning method [42]. In the second approach, we investigated four different CNN models based on the transfer learning method; namely, model A, model B, model C, and model D. Each model is designed based on four popular pre-trained models, namely, VGG16 [43], inceptionV3 [44], efficientNetB0 [45], and DenseNet201 [46], respectively. The purpose of this approach is to re-configure and train only the classification part of each model, while the features extraction part remains the same un-trained, as shown in Figure 7; i.e., the last layers of each pre-trained model are dropped and fine-tuned by two new layers close to the classification layers of the simple model. These two layers are a fully connected layer of 16 hidden neurons with ReLU activation function, a dropout rate of 0.2, a decisive fully connected layer with one neuron, and Sigmoid activation function for a binary classification ($H_{1}$/label 1 for the presence of the PU signal, or $H_{0}$/label 0 for the absence of the PU signal).

Simple CNN model: Includes an input layer, three convolution layers, three max-pooling layers, three dropout layers, one fully connected layer, and an output layer. The input layer takes images of size 128 × 128, and three channels. The three convolution layers used filters of small size, 3 × 3, stride 1, and max pooling layers with filters of 2 × 2 and stride 2. The first convolution layer contains 16 filters, while the second and the third convolution layer contain 64 filters. For all hidden layers, the ReLU activation function is used and a dropout rate of 0.2 is performed. Finally, a fully connected layer with 16 neurons and an output layer with sigmoid activation function are implemented for the final classification decision.Model A: is a combination of the features extraction part of the pre-trained VGG-16 model and the new proposed classification part. The VGG-16 model has a sequential architecture with various number of filters. It includes 13 convolution layers, five pooling layers, three fully connected layers, and an output layer. The input layer takes images of size 224 × 224, and three channels. Each convolution layer used filters of size 3 × 3, stride 1, and the same padding and max pooling layers as the filters of 2 × 2 and stride 2. Convolution layer 1 contains 64 filters, convolution layer 2 contains 128 filters, convolution layer 3 contains 256 filters, and convolution layer 4 and 5 contain 512 filters. For all hidden layers, the ReLU activation function is performed. The first two fully connected layers contain 4096 neurons each, and the third contains 1000. The output layer of VGG-16 used softmax activation function for the final classification decision.Model B: is a combination of the features extraction part of the pre-trained InceptionV3 model and the new proposed classification part. InceptionV3 is an advanced version of the standard InceptionV1 model that was released under the name GoogLeNet in 2014. Compared to the other CNN model, the most important property of the InceptionV3 model is the integration of the inception module, which has a sparsely connected architecture. This module performs multi-convolution at the input with different filters sizes and pooling layers simultaneously, which therefore leads to more complex features extraction. Overall, the InceptionV3 model includes 42 layers arranged under several symmetric and asymmetric components, including convolutions, average pooling, max pooling, concatenations, dropouts, fully connected layers, and an output layer with softmax activation function.Model C: is a combination of the features extraction part of the pre-trained EfficientNetB0 model and the new proposed classification part. The key concept of EfficientNetB0 is the implementation of a new scaling method, unlike the conventional one. This method scales the three dimensions of an input, namely width, depth, and resolution with a compound coefficient; i.e., it carries out the scaling using fixed coefficients. The EfficientNetB0 architecture is organized as a seven mobile-inverted bottleneck convolution, also known as MBConv block, which uses an inverted residual structure for an improved performance of the CNN model. Each block has two inputs: data and arguments. The data inputs are the output data from the previous block, and the argument inputs are a set of attributes such as squeeze ratio, input filters, and expansion ratio.Model D: is a combination of the features extraction part of the pre-trained DenseNet201 model, and the new proposed classification part. DenseNet201 architecture includes four dense blocks using the concept of dense connections between the different layers. To achieve this connection, each layer exploits the feature maps of all previous layers as inputs. The input layer of DenseNet201 takes images with a size of 224 × 224 and three channels. Each two dense blocks are separated by a transition layer, which contains a convolution layer with a filter of 1 × 1, followed by an average-pooling layer with a filter of 2 × 2. At the end, a fully connected layer with 1000 neurons and an output layer with softmax activation function are implemented for the final classification decision.

## 3. Results and Discussion

To evaluate the performance of the proposed approach, we conducted extensive simulations using a computer with an Intel(R) Core(TM) i7-6600U CPU (Central Processing Unit) at 2.60 GHz, 8 GB RAM (Random Access Memory), and Windows 10 64-bit OS (Operating System). The simulation section is divided into two independent parts with different datasets. In the first part, we focused on the performance study of the proposed stacking model-based solution regarding the confidentiality aspect of the sensing data. In the second part, we discussed the performances of different scalogram-based CNN models proposed to improve the decision accuracy of the spectrum sensing process.

### 3.1. Simulation of Stacking Model-Based MUs Detection

In this sub-section, we carried out an extensive quantitative comparison study between the performance results of the individual models (SVM, LR, and NB) and the stacking model based on several performance metrics and in the presence of varying percentages of MUs. For the simulation results, we generated a dataset composed of 2000 examples with five features (Re, SER, Saprsity error, MSC, and MSE) and a binary target (1 for MUs and 0 for SUs). These examples were collected from the experimental results of an original sparse signal ms of length N = 200 samples and randomly placed k = 20 spikes (10% of N). For the compression processes, the sensing matrices $M_{1}$ and $M_{2}$ are generated based on a Chebyshev map. For the recovery process, we used the Bayesian algorithm to speed up the recovery time and to deal with uncertainty in the measurements [47]. For the training process, the dataset is split into 60% for training and 40% for testing.

To measure the prediction performances of the proposed model-based MUs detection, several evaluation metrics are used. These metrics can be classified into two categories. The first category evaluates the model performances based on indirect comparison using the confusion matrix output, which is a square matrix for our case, as we have a binary classification problem [48]. The confusion matrix output includes the following four variables:•True Positive (TP): Malicious users successfully recognized as malicious users.•True Negative (TN): Normal users successfully recognized as normal users.•False Negative (FN): Normal users wrongly recognized as malicious users.•False Positive (FP): Malicious users wrongly recognized as normal users.

Based on these variables, the performance metrics in the first category used in this paper can be defined as follows:-Accuracy: This refers to the percentage of successfully classified nodes over the global number of nodes assessed. It can be defined as
(8)$$\mathrm{Accuracy}=\frac{\mathrm{TP}+\mathrm{TN}}{\mathrm{TP}+\mathrm{TP}+\mathrm{FP}+\mathrm{FN}}$$-Precision: This can be used as a quality metric to evaluate a machine leaning model. It measures the degree of accuracy of the model on positive predictions. It can be described as
(9)$$\mathrm{Precision}=\frac{\mathrm{TP}}{\mathrm{TP}+\mathrm{FP}}$$-Recall: It can be used as a quantity metric to evaluate a machine learning model. It quantifies the total number of correct positive predictions performed from all positive predictions that could have been performed. It can be described as
(10)$$\mathrm{Recall}=\frac{\mathrm{TP}}{\mathrm{TP}+\mathrm{FN}}$$-Authentication Rate (AR): This refers to the rate of the SUs detection. It can be described as
(11)$$\mathrm{AR}=\frac{\mathrm{TN}}{\mathrm{TN}+\mathrm{FN}}$$-Intrusion Rate (IR): This refers to the rate of the MUs misdetection. It can be described as
(12)$$\mathrm{IR}=\frac{\mathrm{FP}}{\mathrm{FP}+\mathrm{TP}}$$The second category evaluates the model performances based on a direct comparison between the true values tested by the model and the final predicted values of the same model. Examples of these evaluation metrics include the Log loss score, Hamming loss score, and the Jaccard score.-Log loss score: Also denoted by cross-entropy loss, it shows the extent to which the prediction probability is approximately close to the related true value. A high log loss score corresponds to a large divergence between the prediction probability and the true value. It can be expressed as
(13)$$L\left(X_{i},y_{i}\right)=-\sum_{c=1}^{C}y_{ic}\times \log\left(P_{ic}\right)$$
where $X_{i}$ denotes the input vector, $y_{i}$ the corresponding target, and $P_{ic}$ is the probability that the *i*th sample is classified in the class c.-Hamming loss score: refers to the amount of incorrect labels relative to the total number of labels. It can be defined as
(14)$$H\left(y,\tilde{y}\right)=\frac{1}{n_{labels}}\sum_{j=1}^{n_{labels}-1}\left(y_{j}\neq \tilde{y_{j}}\right)$$
where $\tilde{y}$ is the predicted value, *y* is the corresponding true value, and $n_{labels}$ presents the number of classes.-Jaccard score: Also called the Jaccard similarity coefficient, it is used to evaluate the similarity between the predicted values and the true values. It is computed as the result of the quotient of the dimension of the intersection by the dimension of the union of two labels. The Jaccard score can be expressed as
(15)$$J\left(y,\tilde{y}\right)=\frac{|y\cap \tilde{y}|}{|y\cup \tilde{y}|}$$
where *y* and $\tilde{y}$ are the true value and the corresponding predicted value, respectively.

Examples of the results are detailed in Table 2 and Figure 8, Figure 9 and Figure 10. Table 2 represents the comparative results of a balanced dataset containing equal proportions of MUs and SUs (50%–50%) ) based on several metrics, namely accuracy, precision, recall, log loss score, Hamming loss score, Jaccard score, and processing time. As one can see, the stacking model achieves 97% accuracy and exceeds the individual models: SVM, LR, and NB, with 70%, 93%, and 80%, respectively. For the precision metric, the four models track almost the same pattern of accuracy performances, with 96% for the stacking model, 66% for the SVM model, 87% for the LR model, and 89% for the NB model. However, the LR and NB models record the highest recall score with 99% for each one, followed by the stacking model with 98%, while the SVM still has a low score value, with 75%. In terms of the log loss and Hamming loss scores, the stacking model has a low value for both metrics, with 0.86 and 0.02, respectively, followed by the NB and LR models, which learn with log loss values of 2.07/2.37 and Hamming loss values of 0.06/0.07, respectively. The SVM model reaches the highest log loss and Hamming loss, with values of 10.2 and 0.3, respectively. For the Jaccard score, the stacking model reaches a higher score, with 94%. The greater the percentage, the more similarity between the sets of real samples and predicted samples. The SVM model performs with a lower Jaccard score of 55%, while the LR and NB models perform roughly similarly, with acceptable values of 87% and 88%. In addition, the required time for processing each model is another evaluation metric that can be exploited for this quantitative comparison. Thus, in terms of processing time, MUs detection takes 0.09 s, 0.05 s, and 0.12 s to be processed with the SVM, LR, and NB models, respectively. Conversely, it takes a longer processing time with the stacking model, due to the multiple tasks executed to provide an accurate prediction with a minimum of errors. To summarize, the proposed stacking model achieves a high accuracy, precision, and Jaccard score, as well as low log loss and Hamming loss scores. Therefore, stacking-based MUs detection can be used as an efficient solution to enhance CRN nodes classification compared to the individual models.

A further alternative for comparing the performance of the different models is via a curve-based comparison, including the Precision-Recall (PR) curve. Figure 8 represents an example of a PR curve, where the precision scores are plotted on the *y*-axis as a function of the recall scores on the *x*-axis. It illustrates the trade-off between precision and recall for various thresholds. The highest precision and recall values are desired to achieve a high model performance; in other words, the appropriate PR curve has the greatest area under curve (AUC). In Figure 8, the stacking model provides the best performance results, with 97.5% of the AUC, followed by the NB model with 94.2% of the AUC, the LR model with 93.3% of the AUC, and SVM with a low value, 70.4% of the AUC.

In addition, to evaluate the robustness of the proposed model in the presence of varying percentages of MUs, we plotted the authentication rate and intrusion rate of the SVM, LR, NB, and stacking-based models against different percentages of malicious users, including 10%, 30%, and 50%. Example of the results are presented in Figure 9 and Figure 10. Figure 9 illustrates the variation of the authentication rate for the different models. It can be observed that the authentication rate increases slightly with the increase in the percentage of MUs for all models with an acceptable rate. Otherwise, all models keep a suitable authentication rate despite the decreasing number of SUs, to reach 98%, 99%, and 99% for the LR, NB, and stacking models, respectively, with only 50% of SUs. The SVM model is an exception, and shows a reduced authentication rate of 80% for the same percentage of MUs. Figure 10 shows the variation of the intrusion rate for the different models. As one can see, the intrusion rate increases progressively with the increase in the percentage of malicious users for all models, which can be considered as acceptable. However, the stacking model shows the low intrusion rate for all percentages than the other models. It reaches only 5% of the intrusion rate with 50% MUs, followed by LR with 12%, NB with 18%, and lastly, SVM with the highest value of 20%. Therefore, the proposed stacking model provides a high authentication rate and a low intrusion rate for the different percentages of malicious users. Thus, we can conclude that the stacking model is still the best solution that gives better performance results; and then it can be selected as a suitable choice for best classifying the CRN nodes, even in the case of datasets with 50% of MUs.

### 3.2. Simulation of a Scalogram-Based CNN Model for Spectrum Sensing

In this sub-section, we performed a comparative analysis between two approaches, the one without transfer learning and the other with transfer learning, to select the suitable model for PU detection. For the first approach, three models of simple CNN were considered, including, simple CNN based on the Adaptive Moment Estimation (Adam) optimizer [49], simple CNN based on the Root Mean Square Propagation (RMSProp) optimizer [50], and simple CNN based on the Stochastic Gradient Descent (SGD) optimizer [51]. For the second approach, four modified pre-trained models according to Section 2.2.2 are considered; namely, the VGG-16-based model A, the InceptionV3-based model B, the EfficientNetB0-based model C, and the DenseNet201-based model D.

For the simulation results, we generated a dataset of 550 sclagoram images, with 300 scalograms presenting occupied signals and 250 scalograms presenting unoccupied signals (noise). For a proof of concept, we considered a band of interest B = [700–800 Hz], a sampling frequency of 8 KHz, and an SNR range from −30 to 30 dB. The spectrum is considered to be occupied if the scalogram shows frequencies in the range 700 to 800 Hz. The spectrum is considered unoccupied (noise) if the scalogram shows either only noise or frequencies in the range out of 700–800 Hz. For the training process, the dataset is split into 70% for training and 30% for testing. For the performance evaluation, accuracy, classification time, the probability of detection ($P_{d}$), and probability of false alarm ($P_{fa}$) are considered as metrics related to the proposed approaches’ evaluation. The accuracy metric is defined according to Equation (Equation 8). The classification time is the time required for training data in the added classification part. The probabilities of detection and false alarm are defined mathematically based on the confusion matrix parameters as follows:(16)$$P_{d}=\frac{N_{TP}}{N_{p}}\;P_{fa}=\frac{N_{FN}}{N_{n}}$$
where $P_{d}$ is the probability that the predicted values are classified into class 1 (the presence of a signal), while the actual values are classified into class 1 (the presence of a signal); $N_{TP}$ is the number of true positives and $N_{p}$ is the total number of selected positives. $P_{fa}$ is the probability that the predicted values are classified into class 1 (a presence of signal), while the actual values are classified into class 0 (an absence of signal). $N_{FN}$ is the number of false negatives and $N_{n}$ is the total number of selected negatives.

Examples of the results are detailed in Table 3 and Figure 11, Figure 12, Figure 13 and Figure 14. Table 3 represents the comparative results in terms of accuracy and classification time between the different proposed CNN models, using a batch size equal to 5 and a number of epochs equal to 15. As one can see, the RMSProp-based simple CNN achieves the highest performance results in terms of both accuracy and time, with 100% training accuracy and testing accuracy, and a faster classification time of 132 s; followed by an Adam-based simple CNN with very close results and a training accuracy of 99.74%, a testing accuracy of 100%, and a classification time of 135 s. The SGD-based simple CNN reaches a training accuracy of 95.55%, a testing accuracy of 98.18%, and a classification time of 137 s. For the transfer learning-based models, the EfficientNetB0-based model C is much faster than the other models (model A, model B, and model D), with a classification time of 165 sec, and it also achieves an acceptable accuracy performance, with 95.06% training accuracy and 96.36% testing accuracy. The VGG16-based model A, InceptionV3-based model B, and DenseNet201-based model D present lower accuracy performances, with 84.90%, 89.87%, and 90.39% of the training accuracy, respectively; and 93.35%, 93.94%, and 96.97% of the testing accuracy, respectively. As well, for the classification time, the three models (model A, model B, and model D) take a long time to train compared to the other models, with 1218 s, 278 s, and 950 s, respectively. Therefore, the proposed simple CNN models give high training and testing accuracies, and a faster classification time.

An alternative way to properly analyze the difference between the models in terms of accuracy is to display its variation as a function of the epochs, as illustrated in Figure 11 and Figure 12. As we can see, both the training and testing accuracies increase proportionally with the number of epochs for both techniques, the simple CNN without transfer learning and the models with transfer learning. Furthermore, it is clear that the CNN models without transfer learning achieve the highest results with a lower number of epochs, and then present a suitable choice for PU detection. For the training accuracy, the CNN model with RMSProp optimizer reaches 100% accuracy in 13 epochs, the CNN model with Adam optimizer reaches 99.74% accuracy in 14 epochs, and the CNN model with SGD optimizer reaches its highest accuracy value of 95.55% in 11 epochs. For the testing accuracy, the CNN model with RMSProp optimizer reaches 100% accuracy in eight epochs, the CNN model with Adam optimizer reaches 100% accuracy in 15 epochs, and the CNN model with SGD optimizer reaches its highest accuracy value of 98.18% in 10 epochs.

Further performance results of our comparative study are presented in Figure 13 and Figure 14, where the probability of detection and false alarm are plotted against the number of epochs. As we can see, the $P_{d}$ increases proportionally with the number of epochs, while the $P_{fa}$ decreases proportionally with the number of epochs for both techniques, the simple CNN models without transfer learning and the models with transfer learning. In addition, the simple CNN models with the RMSProp and Adam optimizers achieve the best performances results in terms of $P_{d}$ and $P_{fa}$. For $P_{d}$ results, the CNN model with both the RMSPprop and Adam optimizers reached 100% of $P_{d}$ in six epochs and nine epochs, respectively. For the $P_{fa}$ results, the CNN model with RMSPprop optimizer met 0% of $P_{fa}$ in eight epochs and the CNN model with Adam optimizer met 12% of $P_{fa}$ in 12 epochs.

Therefore, we can conclude that the significant performance differences between the models trained with transfer learning and the simple CNN models without transfer learning prove that an appropriate design of a sequential CNN model can be a more efficient solution than transfer learning. Furthermore, the CNN model with RMSProp gives the best performance results to perform scalogram-based spectrum sensing processes in CRN, with high accuracy, a high probability of detection, a low probability of false alarm, and a low classification time, with a lower number of epochs.

## 4. Conclusions

In this paper, two new machine learning solutions are proposed for an improved cooperative CRN. The first solution aims at classifying a set of users cooperating in the same network into trusted and malicious users with high accuracy. This solution involves two steps: compressive sensing-based authentication and binary classification based on a stacking model. The first step performs a chaotic compressive sensing technique-based authentication to gather the different features used to build our dataset. The second step performs a binary classification (SU/MU) based on a stacking model with three base classifiers, namely SVM, LR, and NB. The second solution uses the time-frequency representation (2D) of a signal based on scalograms to decide on the occupancy of the PU spectrum (H0 or H1). To implement this solution, several CNN models are compared, including a simple CNN model with three different optimizers, and four modified pre-trained models: VGG16, InceptionV3, EfficientNetB0, and DenseNet20. To review the effectiveness of the proposed stacking model and CNN models, we opted for extensive simulation comparisons. Firstly, the performances of the stacking model-based ensemble learning and the individual learning are compared using several evaluation metrics, including accuracy, precision, recall, log loss score, Hamming score, Jaccard score, processing time, authentication rate, and intrusion rate. Secondly, the performances of the simple CNN models without transfer learning and the modified pre-trained models with transfer learning are compared in terms of accuracy, classification time, probability of detection, and false alarm. Based on the simulation results, we conclude that the proposed stacking model is a promising solution to improve the security issues of cooperative CRN, with a high accuracy of 97%, even in the presence of 50% MUs. As well, the simple CNN model with RMSProp optimizers is the efficient solution for performing a scalogram-based spectrum sensing process in CRN with a high probability of detection, a low probability of false alarm, and a faster classification time. In future work, we will improve the proposed technique for the classification of the PU spectrum. A more extensive study will be performed using a wider and more real-world dataset. In addition, we will apply the proposed concept to more than one SU, with a detailed description of the approach for transmitting the sensing data from multiple SUs to the fusion center, and the technique that will be used to make the final decision. 

## Figures and Tables

**Figure 1 sensors-22-06477-f001:**
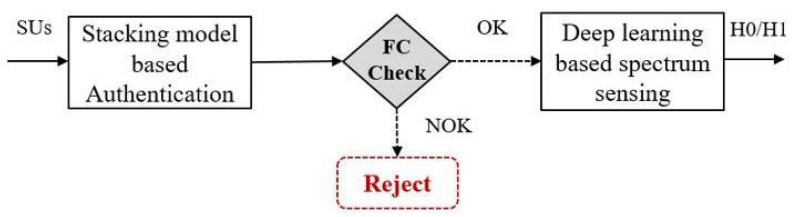
System model blocks.

**Figure 2 sensors-22-06477-f002:**
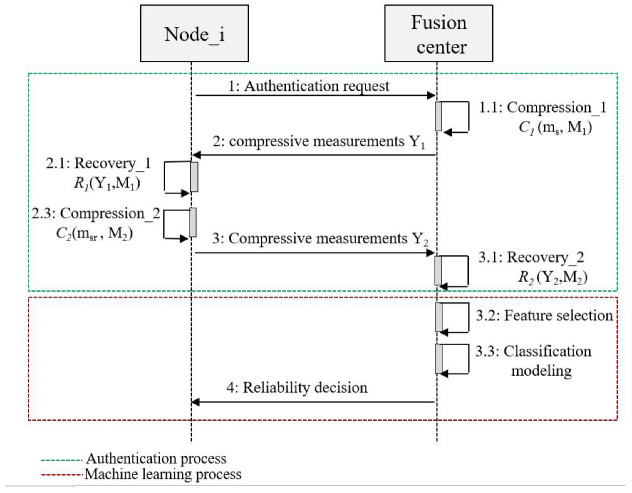
Machine learning processes-based authentication.

**Figure 3 sensors-22-06477-f003:**
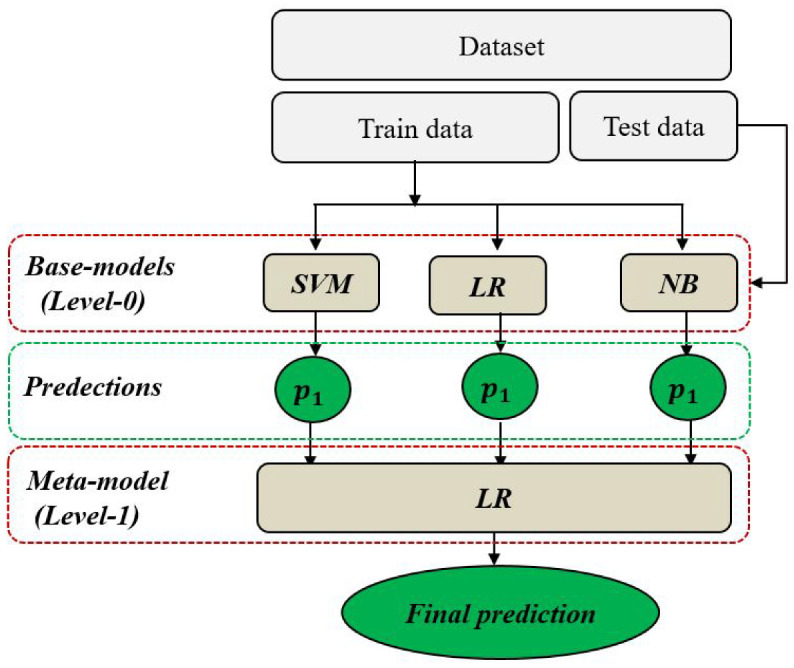
Stacking model with three base models.

**Figure 4 sensors-22-06477-f004:**
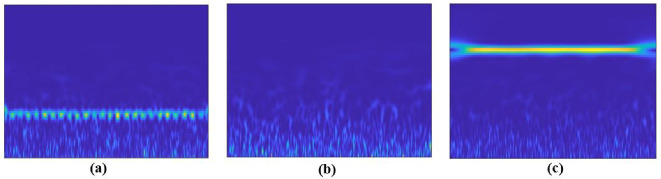
Examples of scalogram: (**a**) Presence of the PU signal in the band of interest; (**b**) absence of PU signal (noise); (**c**) absence of PU signal (presence of frequencies out of the band of interest).

**Figure 5 sensors-22-06477-f005:**
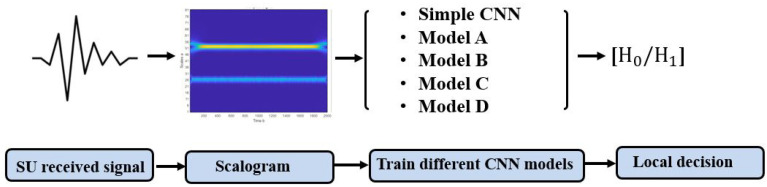
Scalogram-based CNN models processes.

**Figure 6 sensors-22-06477-f006:**
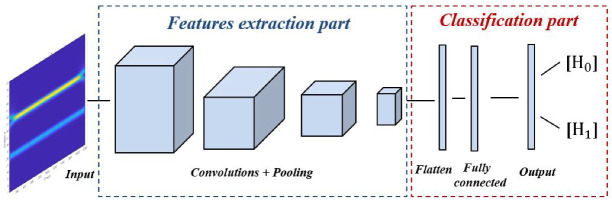
Main parts of CNN models.

**Figure 7 sensors-22-06477-f007:**
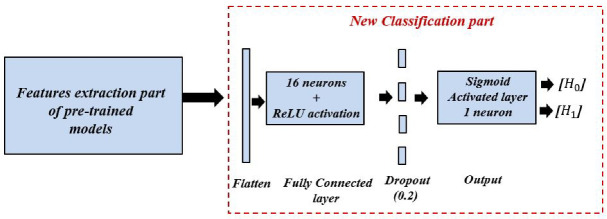
The configuration of the newly added classification part.

**Figure 8 sensors-22-06477-f008:**
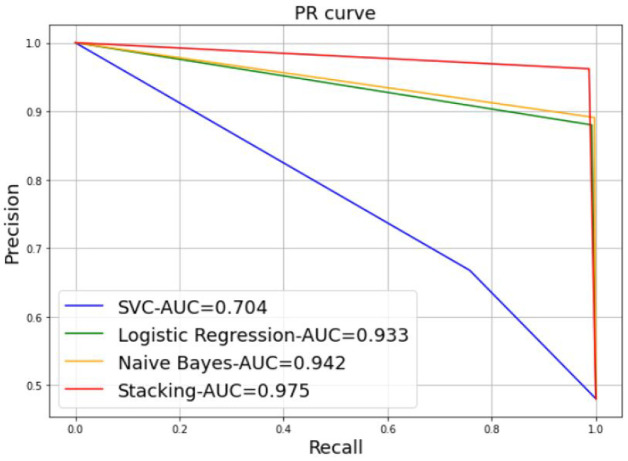
Precision-Recall curve.

**Figure 9 sensors-22-06477-f009:**
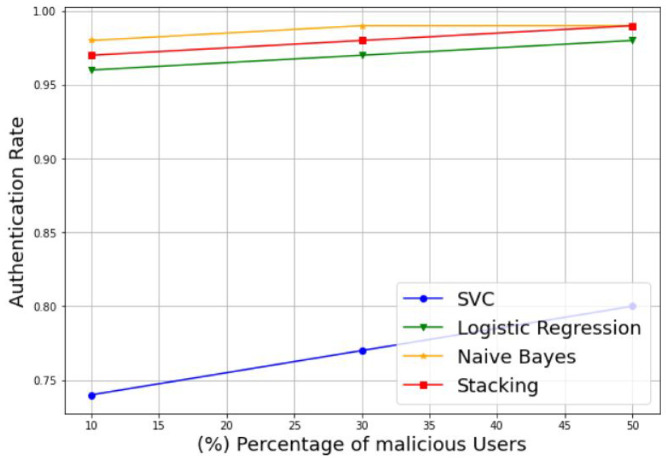
Authentication rate of different percentages of malicious users.

**Figure 10 sensors-22-06477-f010:**
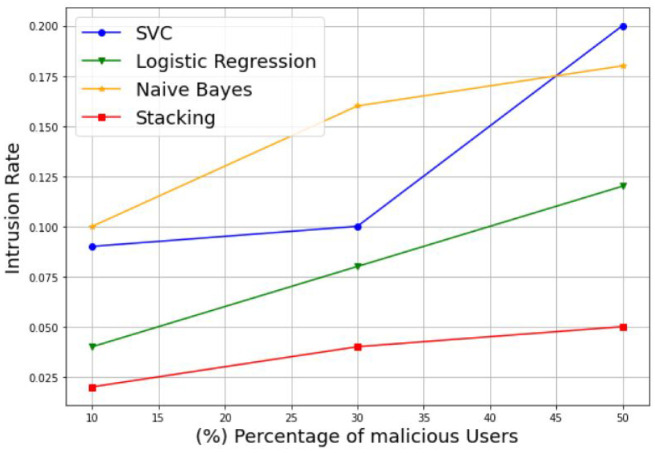
Intrusion rate of different percentages of malicious users.

**Figure 11 sensors-22-06477-f011:**
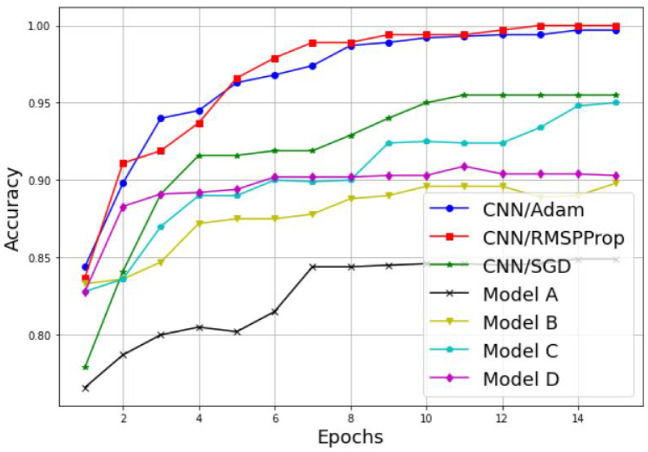
Training accuracy comparison results between different CNN models.

**Figure 12 sensors-22-06477-f012:**
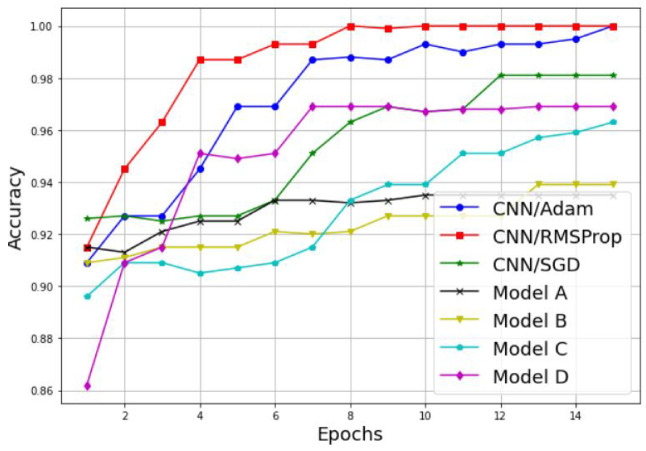
Testing accuracy comparison results between different CNN models.

**Figure 13 sensors-22-06477-f013:**
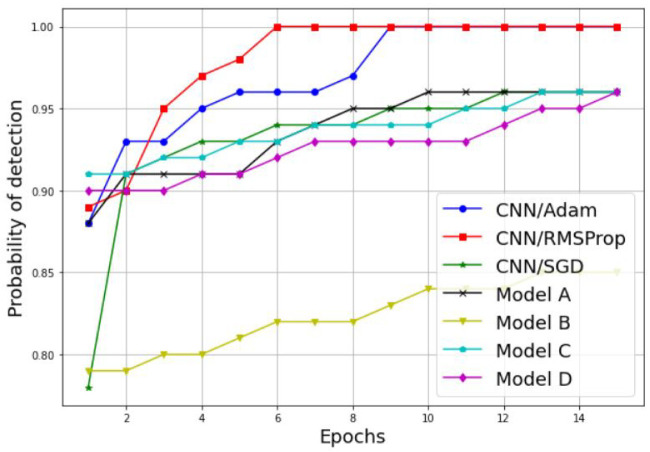
Probability of detection comparison between different CNN models.

**Figure 14 sensors-22-06477-f014:**
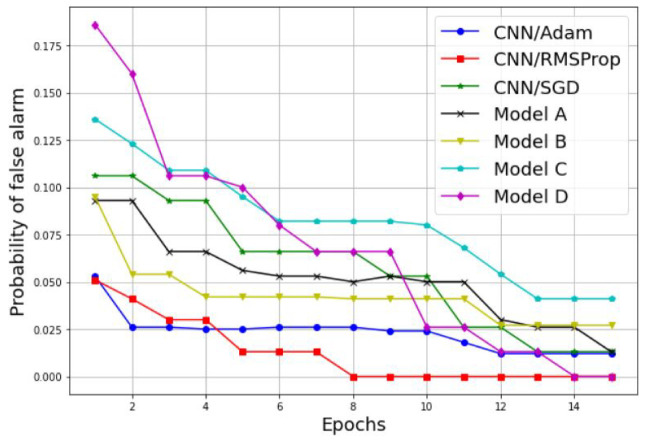
Probability of false alarm comparison between different CNN models.

**Table 1 sensors-22-06477-t001:** Comparison of existing works based on machine learning.

References	Approach	Machine Learning Algorithms	Features	Evaluation Metrics
Ref. [14]	MU detection	SVM	Energy feature	Accuracy/ROC/
				$P_{d}$ and $P_{fa}$
Ref. [15]	MU detection	LR/KNN/SVM	Energy feature	Accuracy
				Sum rate
				Network lifetime
Ref. [16]	MU detection	LR/KNN/SVM/	SNR feature	Accuracy/Recall
		LDA/DTC/NB	Entropy feature	Precision/F-score
			PU state	
Ref. [17]	MU detection	Feed-forward network	Energy decay rate	ROC
			Gradient vectors	Area under ROC
Ref. [18]	PU detection	SVM/KNN/DCT/	Energy feature	Accuracy
		NB/LR		$P_{d}$/$P_{fa}$
Ref. [19]	PU detection	DTC/SVM/KNN/	Energy feature	Accuracy/Pd
		Weighted ensemble	Wavelet feature	ROC/Training time
			SNR feature	Predection speed
Ref. [20]	PU detection	CNN	Cyclostationary feature	Loss function
			Energy feature	$P_{d}$
Ref. [21]	PU detection	CNN/LSTM/	Automatic features	ROC
		Fully connected neural network	extraction	$P_{d}$

**Table 2 sensors-22-06477-t002:** Quantitative comparison of the individual models and stacking model.

Models	SVM	LR	NB	Stacking
Accuracy	0.70	0.93	0.94	0.97
Precision	0.66	0.87	0.89	0.96
Recall	0.75	0.99	0.99	0.98
Log loss	10.2	2.37	2.07	0.86
Hamming loss	0.3	0.07	0.06	0.02
Jaccard score	0.55	0.87	0.88	0.94
Processing time	0.09	0.05	0.12	0.2

**Table 3 sensors-22-06477-t003:** Comparison results between different CNN models.

CNN Model	Optimizer	Transfer Learning	Training Accuracy	Testing Accuracy	Classification Time
Simple CNN	Adam	No	99.74	100	135 s
Simple CNN	RMSProp	No	100	100	132 s
Simple CNN	SGD	No	95.55	98.18	137 s
Model A	Adam	Yes	84.90	93.35	1218 s
Model B	Adam	Yes	89.87	93.94	278 s
Model C	Adam	Yes	95.06	96.36	165 s
Model D	Adam	Yes	90.39	96.97	950 s

## Data Availability

Not applicable.

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
