# Peer review of "A Novel Prediction Model for Malicious Users Detection and Spectrum Sensing Based on Stacking and Deep Learning"

_sensors, 2022, doi:10.3390/s22176477_

Round 1
Reviewer 1 Report
In this paper, the authors proposed a new stacking model-based malicious users detection for users classification (SU/MU) and a novel deep learning technique using scalogram images as inputs for primary user spectrum classification (H0/H1). The illustrations are informative to readers. However, some drawbacks exist in the current form of the manuscript, and I have some concerns that need to be addressed by the authors as follows:
- The current title seems too general. It should be rewritten by a more specific one that highlights the manuscript's contribution.
- In line 34, the authors claimed that “Despite the key benefits of cooperative networks, they are susceptible to potential attacks by malicious users (MUs) causing undesirable interferences between SUs and PUs and consequently, reducing the decision accuracy of the spectrum sensing process”. Indeed, the SUs and PUs in CRNs generally do not cooperate for spectrum sensing. Please justify this.
- The mentioned references are informative. In my opinion, however, it is better to provide a comparative table in the introduction to illustrate the difference between the proposed schemes and other classification and deep learning techniques regarding spectrum sensing.
- In Algorithm, what do the X_i and y_i represent? For my understanding, X_i represents the sensing signal, and y_i is the local spectrum result (known as labels of input X_i) which shows the PUs occupancy (H0/H1). If so, how can the classifiers in base-models (SVM, LR, and NB) classify the type of user (SU/MU) at the output? Besides, I recommend using the indent in the loops of the pseudo-code algorithm.
- In Section 2.2.2, when is the transfer learning used in models A, B, C, and D? If possible, the impact of transfer learning on detection performance should be discussed.
- In the simulation section, each plotted line in the figures is required to be added with a marker to differentiate one line from the others.
- In table 2, the results show that the optimizer RMSProp offers the best performance compared to the others. However, the training accuracy and testing accuracy are perfect at 100%. The explanation for this should be mentioned.
- Future work is needed in the conclusion section.
- Typing and grammar errors are easily recognized throughout the manuscript. I believe the authors can pay more attention to improving it with a better version.
Author Response
August 16, 2022
Dear Editor(s) and Reviewer(s),
We highly appreciate the constructive review of our manuscript No. Sensors-1852554 with detailed and valuable comments from the reviewers. We have comprehensively addressed the concerns raised by the area editor and reviewers. In addition, with the intent to improve the quality of the manuscript, several required changes are made to the previous submission. Please find listed below our responses to the reviewers’ comments. We hope that our responses and the revised manuscript will be satisfactory.
Please note: Authors’ responses are reported in blue. The corresponding amendments to the manuscript are highlighted in yellow. We did not highlight the other revisions in order to keep track of the reviewers’ comments.
We are grateful to the editors for re-considering our manuscript for this esteemed journal. We look forward to receiving a positive response from your side.
Sincerely,
Salma Benazzouza, et al.
GREENTIC/RITM Laboratory, CED Engineering Sciences
Hassan II University
salma.benazzouza@ensem.ac.ma

Reviewer 2 Report
In this paper, the authors proposed two new machine learning solutions to improved cooperative CRN. I have the following questions for the authors before the possible publication in Sensors.
1. Please define a malicious user. If a user is identified as a MU, does it mean that the user will be treated as a MU forever?
2. If a CR is with low quality sensing data, perhaps because it is far from a Primary use, is it possible to be identified as a MU.
3. How to acquire the data set for training to classify SU/MU? And who is in charge of this important task.
4. Where to performance cooperate sensing, at a fusion center? How about the transmission for sensing data and final decision?
5. In Fig 14, it seems that there is no change from epoch 2 to 10. Why? Is it just one training result?
6. Have the authors considered the PUs situation, such as SNR of a primary signal?
7. Have the authors considered the dynamic changes of occupancy of Primary channels. Do the two learning techniques can handle this issue?
8. It is necessary to declare the cost of two steps learning, such as delay, energy consumption, or complexity.
Author Response

(The authors gave the same response as above.)

Round 2
Reviewer 1 Report
Thank you for the authors' hard-working study and revision. Most of the comments of the reviewers are reflected with well-addressed responses.
However, some minor typos still exist. Please check it carefully to let the paper be ready for publication.